# Towards Human-Preferences Chinese Rewriting Evaluation: Prompt-Based Scoring with Large Language Models

## Abstract

Sentence rewriting is a core task in natural language processing, encompassing paraphrasing, translation, and summarization. Despite its importance, existing evaluation metrics often rely on superficial similarity measures (e.g., BLEU, ROUGE), which fail to capture deep semantic fidelity. In this work, we propose a principled, multi-dimensional framework for evaluating rewriting quality based on semantic consistency, syntactic structure, lexical variation, and stylistic fidelity. We design a prompt-based scoring method with the QWQ-32B language model, achieving a Spearman correlation of $\rho = 0.6121$ with human judgments, which is comparable to inter-human agreement ($\rho = 0.6076$). We further benchmark popular rewriting strategies using this metric and introduce a multiround generation pipeline that improves rewriting quality by 9.66%. Our results show that large language models, when paired with structured evaluation and guidance, can robustly assess and generate high-quality rewrites.

## 1 Introduction

Sentence rewriting (e.g., paraphrasing, translation, or summarization) is a fundamental task in NLP that requires restating content without changing its meaning Chen & Bansal (2018); Zhang et al. (2019b); Zhang & Litman (2014). Examples include translating a sentence into another language or condensing a passage into a summary. In all cases, the semantic information must remain consistent before and after rewriting Shen et al. (2022). However, the NLP literature lacks a standard, quantifiable metric for rewriting quality Chen & Dolan (2011). Early approaches treated rewriting as a similarity task: for instance, BLEU Papineni et al. (2002) and ROUGE Lin (2004) (n-gram overlap) or Jaccard and TF–IDF Ramos et al. (2003) measures count shared words and order. Word-embedding methods (Word2Vec, GloVe) and contextual models (BERT) embed words/sentences into vector spaces, with similarity computed by cosine or Euclidean distance Devlin et al. (2019); Reimers & Gurevych (2019). Likewise, neural models such as Siamese networks have been used for semantic matching. For example, Mueller and Thyagarajan (2016) applied a Siamese LSTM Bao et al. (2018) to sentence pairs to learn semantic similarity Mueller & Thyagarajan (2016), and Shi et al. (2020) used a Siamese CNN Leal-Taixé et al. (2016) for Chinese sentence similarity Shi et al. (2020).

Despite these advances, traditional similarity metrics often fail for rewriting: they reward superficial overlap rather than true meaning preservation. Chen and Dolan (2011) noted that a paraphrase metric like BLEU would be maximized by simply copying the input Chen & Dolan (2011). For instance, "I like to eat apples" and "I do not like to eat apples" share many words, but clearly differ in meaning; simple n-gram or Jaccard scores would not catch this semantic flip. In practice, practitioners crudely filter paraphrases by a BLEU range (e.g. 0.6–0.8) based on experience. More recent large language models (LLMs) have deeper semantic understanding; models like GPT-4 can achieve human-level performance on many benchmarks Achiam et al. (2023), suggesting they could better evaluate semantic fidelity. However, using LLMs for rewriting still requires a clear definition of the task and scoring criteria. LLMs also exhibit "language inertia" (tendency to keep output similar to input), so naïve prompting often yields rewrites too close to the original.

Figure 1: Overview of the main evaluation pipeline. The process begins with human-written inputs and proceeds through rewrite generation, human annotation, and metric-based evaluation.

Therefore, we introduce a clear definition and metric for high-fidelity rewriting. We propose evaluation criteria across multiple dimensions: semantic consistency, syntactic structure, lexical variation, and style length. We design prompts to elicit a "rewrite score" from LLMs that correlates with human judgments. In experiments, our QWQ-32B Yang et al. (2025) scoring prompt achieves Spearman $\rho = 0.593$ with human ratings on rewriting. We also benchmark several state-of-the-art LMs on rewriting and fine-tune a rewriting model using data filtered by our score, yielding a 14.95% improvement in rewriting quality.

In summary, our contributions are:

- We analyze the sentence rewriting task and propose a composite evaluation metric consisting of four dimensions: semantic similarity, word order, lexical substitution, and length/style.
- Through extensive experimentation and model tuning, we design a rewriting scoring prompt with explicit evaluation standards. Using the QWQ-32B model, our scoring method achieves a Spearman correlation coefficient of $\rho = 0.6121$ with human ratings, which is comparable to inter-human agreement ($\rho = 0.6076$).
- We systematically evaluate multiple state-of-the-art models and various rewriting generation methods. Based on these evaluations, we propose a multi-step rewriting framework that stably generates high-quality rewritten texts, significantly enhancing the consistency and reliability of rewriting outcomes.

## 2 RELATED WORK

Automatic evaluation for paraphrase or rewriting has been studied only sporadically. Traditional MT-style metrics (BLEU Papineni et al. (2002), METEOR Banerjee & Lavie (2005), etc.) and text-similarity measures (TF–IDF Ramos et al. (2003), edit distance Friendly (2019), Jaccard Niwattanakul et al. (2013)) have been applied, but they inherently reward overlap rather than novel phrasing. Chen and Dolan proposed the PINC (Paraphrase In N-gram Changes) score, which measures the proportion of n-grams in the candidate not present in the source (the inverse of BLEU) to capture lexical novelty Chen & Dolan (2011); Cavnar et al. (1994). They found that combining BLEU (to assess adequacy) with PINC (to assess difference) correlates with human judgments of paraphrase quality. More recently, Shen et al. (2022) showed that good paraphrases follow two criteria: semantic similarity and lexical divergence. They proposed ParaScore, a metric that explicitly models these aspects by combining BERT-based similarity with a divergence component Shen et al. (2022). ParaScore significantly outperforms previous metrics on paraphrase-generation benchmarks.

Other work has explored embedding-based and neural metrics. For example, BERTScore Zhang et al. (2019a) computes semantic overlap using contextual embeddings, and SBERT Reimers & Gurevych (2019); Li et al. (2023) learns sentence embeddings via a BERT network. These methods

capture semantic relatedness beyond surface overlap. However, even strong semantic metrics like BERTScore are not specifically designed to encourage paraphrastic diversity. Siamese neural network architectures (CNNs Guo et al. (2019) or LSTMs Yu et al. (2019)) are commonly used for sentence similarity tasks, but again they mainly quantify closeness, not rewriting fidelity.

With the advent of LLMs, new evaluation paradigms emerge. Models such as GPT-4 Achiam et al. (2023), DeepSeek Liu et al. (2024), GLM-4 GLM et al. (2024) exhibit near-human understanding and can compare sentence meanings directly. Some work has begun using LLMs as reference-free metrics (e.g. via prompting or fine-tuning), but this area is still nascent. Overall, prior metrics have not fully addressed the rewriting task: either they focus on n-gram matching or they fail to balance semantic consistency with lexical novelty. Our work builds on these insights by defining a specialized rewriting metric and prompting strategy that leverages LLMs for high-fidelity sentence rewriting Liu & Mozafari (2024); Shu et al. (2024).

## 3 METHOD

### 3.1 HUMAN EVALUATION AND ANNOTATION PROTOCOL

To establish a reliable reference for evaluating sentence rewriting quality, we first collected human annotations. All rewriting tasks were conducted in Chinese, and all annotators were native Chinese speakers. The English text below is a translation for readability; the complete original Chinese instructions are provided in the appendix.

Two linguistic experts independently rated 200 sentence rewriting pairs based on the following basic and intentionally underspecified criteria:

1. The rewritten sentence should be as different from the original as possible in surface form.

2. The rewritten sentence must preserve the semantic meaning of the original.

3. Each sentence pair is scored on a 0–5 integer scale.

| Rewrite the quality scoring criteria |
|---|
| Requirement 1: Under the premise of maintaining the original meaning unchanged, express in different words of the original text. Do not add or delete any content that would seriously affect the meaning. |
| Requirement 2: Under the premise of ensuring logical correctness and smooth word order, change the sequence and structural relationship of the appearance of entities, phrases, and concepts. The smaller the amplitude of the sequential transformation, the lower the score. |
| Requirement 3: Under the premise of ensuring that special entities, references, explanations, and specific statements remain unchanged, use synonyms for substitution. The fewer synonyms are replaced, the lower the score will be. |
| Requirement 4: Maintain consistency with the original language style, with reasonable variations in word count and length. The greater the gap in language styles, the lower the score. |
| Requirement 5: The rewriting of the comprehensive score should be very strict, taking into account the above four requirements simultaneously. Under the premise of ensuring semantic consistency (Requirement 1), the greater the variation in (Requirement 2,3), the better, and the closer the word count, the better (Requirement 4). If there are a large number of long fields that are completely copied, 0 points will be given. |
| The scoring criteria are as follows: the minimum score is 0 and the full score is 5. The scoring should be as strict as possible. |

Table 1: Multidimensional quality criteria for sentence rewriting: Thresholds and scoring rules for semantic fidelity (Req1), syntactic diversity (Req2), lexical variation (Req3), and stylistic alignment (Req4).

Inter-rater agreement analysis revealed moderate correlation: the Spearman correlation was 0.6076, exact agreement was 34%, within $\pm 1$ score difference was 84.5%, and within $\pm 2$ was 96%. While perfect agreement was limited, the high $\pm 1$ accuracy suggests that the annotators shared a generally consistent understanding of the task, despite some variance in fine-grained judgments.

To further improve annotation reliability, we designed a structured evaluation framework grounded in four explicit dimensions: (1) semantic consistency, (2) syntactic structure, (3) synonym substitution, and (4) stylistic fidelity and sentence length. The refined guidelines are outlined in Table 1.

Using these criteria, we re-annotated 730 samples produced by human rewriting. These 730 samples are used for further feature analysis and represent the ability of humans to rewrite.

## 3.2 TRADITIONAL EVALUATION METRICS

Traditional evaluation of rewriting quality often relies on heuristic similarity measures or rough thresholds, lacking fine-grained interpretability.

| Metric | Level | Semantic | Advantages | Disadvantages | Use Case |
|---|---|---|---|---|---|
| Jaccard | Word | × | Simple, good for short text | Ignores word order | Keyword matching |
| TF-IDF | Word | × | Reduces common word weight | No synonym handling | Document retrieval |
| BLEU | n-gram | × | MT industry standard | Unfriendly to short text | Machine translation |
| ROUGE-L | LCS | × | Focuses on coherence | Biased to long text | Text summarization |
| Word2Vec | Word vec. | ✓ | Deep word-based matching | High complexity | Short text matching |
| Sentence-BERT | Sentence | ✓ | Cross-task general | Needs GPU | Semantic search |
| Jaro-Winkler | Char | × | Good for names/addresses | Poor for long text | Entity alignment |
| TTR | Word | × | Simple & effective | Length-dependent | Writing analysis |
| Keyword | Word | × | Intuitive | Needs good extraction | Content moderation |
| Punctuation | Symbol | × | Detects fluency | Needs other metrics | AI text detection |

Table 2: Comparison of common text similarity metrics by granularity, semantic awareness, strengths, weaknesses, and typical use cases. Traditional metrics are effective in certain contexts but often fail to capture semantic-level rewriting, motivating more context-sensitive evaluation.

Table 2 summarizes commonly used text similarity metrics adapted for sentence rewriting. These range from character-level edit distances to sentence-level embeddings, differing in their ability to capture semantic meaning.

Surface-based metrics such as BLEU, ROUGE-L, and TF-IDF emphasize n-gram or frequency overlap but often miss deeper semantic shifts. Jaccard and TTR highlight lexical variation while ignoring context. In contrast, embedding-based metrics like Word2Vec and Sentence-BERT capture semantic similarity more effectively, though at higher computational cost.

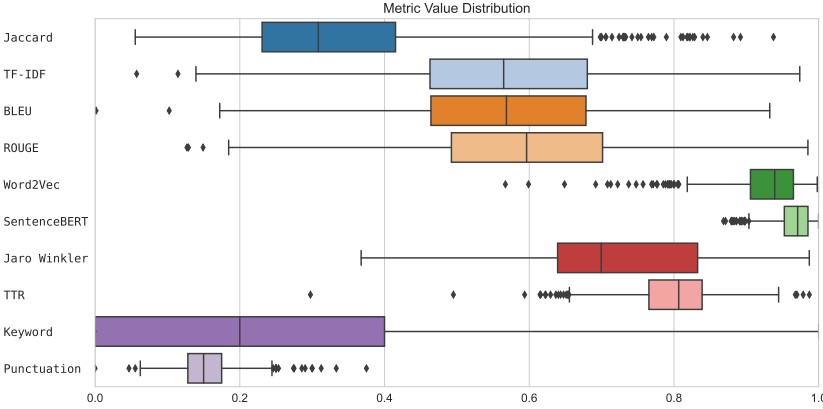

Figure 2: Distribution of traditional text similarity and rewriting metrics. Semantic metrics (e.g., `SentenceBERT`, `Word2Vec`) show higher values with less variance, while surface-level metrics (e.g., `Jaccard`, `TF-IDF`, `Punctuation`) exhibit lower medians and greater spread.

To examine their behavior in rewriting, we analyzed 10 metrics, including lexical overlap (Jaccard, TF-IDF, ROUGE-L), edit-based (Jaro-Winkler), embedding-based (Word2Vec, Sentence-BERT), and stylistic features (TTR, punctuation ratio). Distributions across our dataset (Figure 2) show that embedding-based metrics cluster at higher values, indicating stronger semantic alignment, while surface metrics display wider dispersion and lower medians.

Overall, although widely applied, these metrics were not designed for rewriting tasks that involve intertwined lexical, syntactic, and semantic changes. This motivates our further analysis in Section 4, where we study their correlation with human-rated rewrite quality.

### 3.3 LARGE LANGUAGE MODEL-BASED EVALUATION AND GENERATION

Recent large language models (LLMs) can be broadly categorized into reasoning-intensive models (e.g., DeepSeek-R1) and non-reasoning models (e.g., GPT-4o). Although structurally similar, their output behaviors differ, as reasoning models often perform intermediate thinking before reaching a conclusion.

To explore LLM capabilities in rewriting tasks, we conduct experiments across a range of open-source models of different scales and families, including LLaMA, Qwen, DeepSeek, and GLM.

#### 3.3.1 REWRITING EVALUATION PROMPTS

Prompt design significantly influences LLM output quality. We experimented with four levels of instruction detail for rewriting evaluation:

| Prompt Type | Prompt | Output Format |
|---|---|---|
| None | Appendix 11 | Free-form judgment |
| Only | Appendix 10 | Overall score |
| Only-Reason | Appendix 10 | Overall score + justification |
| Multi-Reason | Appendix 10 | Each score + justification |

Table 3: Prompt formats used for LLM-based evaluation.

Table 3 summarizes the four prompt types used in our evaluation experiments. The **None** setting provides no explicit instruction, relying on the model's default behavior. The **Only** prompt asks for an overall score, while **Only-Reason** adds a brief justification to the score. The most detailed, **Multi-Reason**, requests separate scores for multiple criteria, each with its own explanation. These prompt types allow us to assess how varying levels of instruction influence the alignment between LLM and human evaluations.

#### 3.3.2 REWRITING GENERATION PROMPTS

To generate diverse rewriting examples, we tested four generation strategies using different prompts:

| Prompt Type | Description |
|---|---|
| Back Translation | Translate to another language and back to generate variation. |
| Summarize | Summarize original, then regenerate to a target length. |
| Rewrite | Simply prompt "Please rewrite the following sentence." |
| Guided Rewrite | Rewrite according to criteria in Table 1. |

Table 4: Rewriting generation strategies used in our experiments.

## 4 EXPERIMENTS

### 4.1 EVALUATION OF TRADITIONAL METRICS

As described in Section 3.1, we constructed a human-labeled dataset of 730 rewritten sentence pairs, each rated on a Comprehensive Rewriting Evaluation (CRE) scale from 0 to 5 based on rewriting quality. Traditional evaluation methods for rewriting often rely on empirical thresholds, such as selecting samples with BLEU scores within a certain range, but whether this approach truly filters high-quality rewrites remains an open question. We address this issue through comprehensive quantitative analysis.

As illustrated in Figure 3, we analyzed ten different traditional evaluation metrics as described in Section 3.2. Each curve represents the distribution of rewrite samples under a specific human rating. Notably, metrics such as TTR (Type-Token Ratio) and punctuation ratio exhibit minimal distributional differences across ratings, indicating their limited discriminative power. In contrast, metrics such as BLEU, ROUGE, TF-IDF, Jaccard, Jaro-Winkler, Word2Vec, and Sentence-BERT show

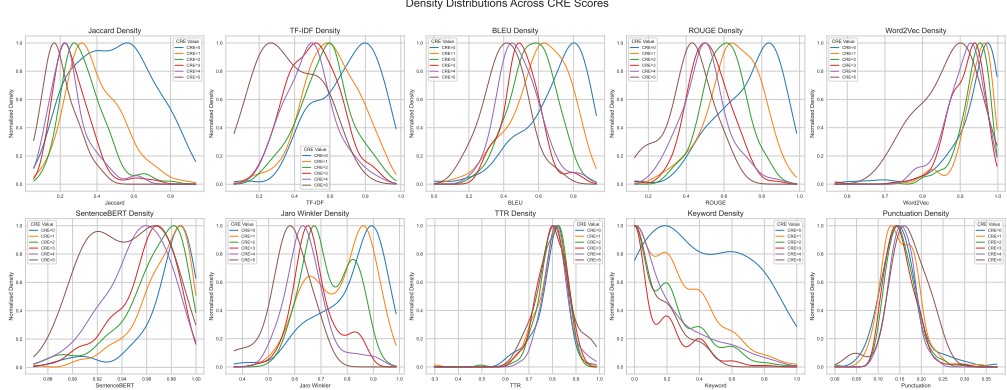

Figure 3: Score distributions across different traditional evaluation metrics. Colors represent human-assigned quality scores from 0 to 5. The x-axis denotes metric values; the y-axis indicates normalized distribution density.

more distinguishable trends, with distribution peaks shifting gradually according to human-assigned quality scores.

For example, in the BLEU score distribution, we observe that BLEU values around 0.6 tend to be associated with low-quality rewrites (CRE scores of 1–2), while scores below 0.6 are more frequently associated with higher-quality rewrites (CRE scores of 3–5). This implies that empirical thresholds like BLEU $\in$ [0.2, 0.5] may effectively filter out poor-quality samples. However, this filtering remains coarse-grained and cannot prevent a significant number of false positives, i.e., retaining low-quality rewrites.

### 4.1.1 Metric Score Trends Across Rewrite Quality Levels

To examine how traditional metrics align with human rewrite quality, Figure 4 shows average trends across CRE levels (0-5), with means, medians, and confidence intervals. Most metrics exhibit a downward slope as quality increases, reflecting that higher-quality rewrites deviate more from the source. However, variance grows at extreme CRE values, suggesting instability and outliers.

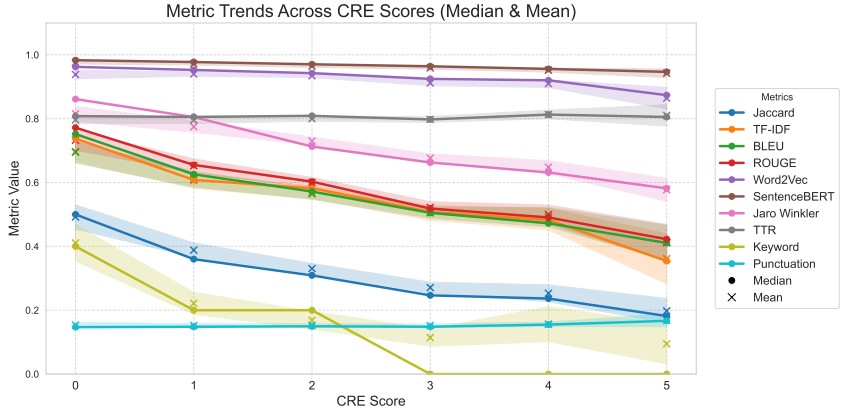

Figure 4: Trend of evaluation scores across human rewrite quality levels (CRE 0–5). Circles denote medians; X markers denote means; shaded areas represent confidence intervals.

Figure 3 further compares score distributions across CRE levels. Metrics like BLEU, ROUGE, Jaccard, and embeddings (Word2Vec, Sentence-BERT) show partial separation, indicating some discriminative ability. In contrast, TTR and Punctuation Ratio display nearly overlapping curves, confirming limited usefulness. While heuristic thresholds (e.g., BLEU ranges) can filter out some poor rewrites, they also exclude many good ones, leading to high false positives/negatives.

Overall, traditional metrics provide only coarse signals: they may distinguish extremes but fail to robustly capture semantic quality. This highlights the need for evaluation methods grounded in meaning representation, such as LLM-based scoring.

## 4.2 EVALUATION OF LLM-BASED METRICS

### 4.2.1 IMPACT OF MODEL SIZE AND INSTRUCTION METHOD ON EVALUATION PERFORMANCE

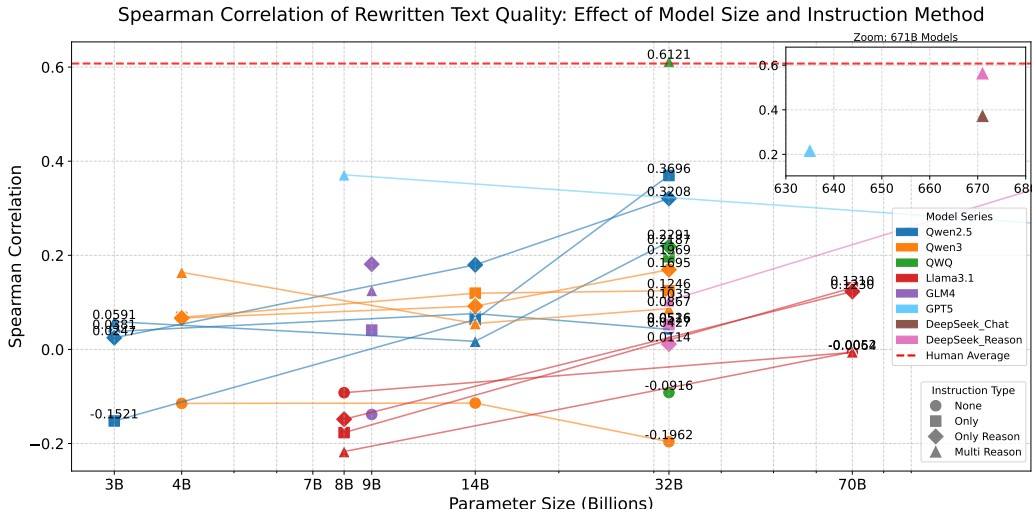

Figure 5: Spearman correlation coefficients between LLM-generated scores and human ratings. The red dashed line denotes inter-annotator agreement, serving as a human-level benchmark.

Figure 5 shows Spearman correlations between LLM-generated rewrite quality scores and human annotations. The red dashed line indicates inter-annotator agreement ($\rho = 0.6076$) as a human-level benchmark. Correlation generally improves with model size: Qwen2.5 rises from $\rho = 0.0591$ (3B) to $\rho = 0.2291$ (32B), and QWQ-32B achieves $\rho = 0.6121$, slightly surpassing human agreement.

Prompting strategy also affects performance. For Qwen2.5-32B, simple prompts perform well, while complex "multi reason" prompts may reduce correlation. Qwen3 and LLaMA3.1 show weaker and inconsistent results, with some large variants even producing negative correlations. DeepSeek-Reason-671B under "multi reason" prompting reaches $\rho = 0.5663$, highlighting the benefit of reasoning-focused instruction tuning.

Overall, instruction tuning is critical for aligning outputs with human judgments. QWQ-32B's strong performance with "multi reason" prompts versus its non-instruct variant ($\rho = -0.0916$) illustrates this effect. While reasoning prompts help, overly complex instructions can degrade performance. These findings emphasize that robust, human-aligned evaluation depends on both model scale and carefully designed prompting.

### 4.2.2 ANALYSIS OF LLM EVALUATION RESULTS

Table 5 presents key trends in model evaluation effectiveness. The inter-annotator Spearman correlation ($\rho = 0.6076$) serves as a human-level benchmark. Notably, **QWQ-32B-Multi-Reason** slightly surpasses this with $\rho = 0.6121$, and **DeepSeek-Reason-Multi-Reason** closely follows at $\rho = 0.5754$, indicating that large models with well-designed prompts can match or exceed human consistency in rewrite evaluation.

Exact match rates further support this: QWQ-32B-MR achieves 47.51% exact match, outperforming both individual annotators (34.00%) and all other models. DeepSeek-Reason-MR achieves 44.86%, with the highest accuracy under tolerant criteria ($\pm1$: 82.70%, $\pm2$: 95.93%). Appendix 7 shows

| Model | Size | Spearman | MAE | RMSE | Kendall | Exact % | ±1 % | ±2 % |
|---|---|---|---|---|---|---|---|---|
| Human Average | - | 0.6076 | 0.8550 | 1.1511 | 0.5192 | 34.00 | 84.50 | 96.00 |
| Qwen2.5 $_N$ | 32B | 0.0427 | 1.7525 | 2.0726 | 0.0385 | 12.75 | 46.75 | 72.00 |
| Qwen2.5 $_N$ | 32B | 0.0427 | 1.7525 | 2.0726 | 0.0385 | 12.75 | 46.75 | 72.00 |
| Qwen2.5 $_{OR}$ | 32B | 0.3208 | 0.9675 | 1.2339 | 0.2823 | 29.00 | 77.00 | 97.25 |
| Qwen2.5 $_{MR}$ | 32B | 0.2291 | 1.2925 | 1.6054 | 0.1969 | 22.50 | 61.25 | 88.00 |
| QWQ $_N$ | 32B | -0.0916 | 2.3225 | 2.6848 | -0.0794 | 7.75 | 31.50 | 56.50 |
| QWQ $_O$ | 32B | 0.1969 | 1.3625 | 1.7017 | 0.1688 | 21.75 | 58.75 | 85.50 |
| QWQ $_{OR}$ | 32B | 0.2187 | 1.1325 | 1.5133 | 0.1880 | 30.75 | 67.75 | 90.25 |
| QWQ $_{MR}$ | 32B | 0.6121 | 0.8203 | 1.2728 | 0.5493 | 47.51 | 79.61 | 93.51 |
| GPT-5-mini $_{MR}$ | ≈ 8B | 0.3710 | 1.0575 | 1.3648 | 0.3105 | 28.00 | 72.75 | 93.75 |
| GPT-5 $_{MR}$ | ≈ 635B | 0.2185 | 1.1125 | 1.4494 | 0.1827 | 27.50 | 70.25 | 92.50 |
| DSeek-C $_{MR}$ | 671B | 0.3751 | 1.0364 | 1.4376 | 0.3277 | 34.77 | 72.78 | 90.61 |
| DSeek-R $_{MR}$ | 671B | 0.5663 | 0.7701 | 1.1243 | 0.5021 | 44.86 | 82.70 | 95.93 |

Table 5: Comparison of model-generated scores with human ratings (including inter-rater agreement). Model name suffixes indicate: N = None, O = Only, OR = Only Reason , MR = Multi-Reason in Table 3. DSeek-C = DeepSeek-Chat model, DSeek-R = DeepSeek-Reason model. Bold values indicate the best performing metrics.

this model aligns more with Expert 1 (64.52%) than Expert 2 (30.50%), though ±1 agreement is comparable (86.71% vs. 72.50%), suggesting robustness to minor scoring deviations.

Incorporating reasoning into prompts generally improves performance. For example, Qwen2.5-32B improves from 65.75% (±1) and 93.25% (±2) in the original setting to 77.00% and 97.25% with reasoning. However, multi-reasoning is not always better; for Qwen2.5-32B, the single-reasoning variant outperforms the MR version, suggesting that overly complex prompts may introduce noise.

## 4.3 ANALYSIS OF LLM-BASED REWRITING METHODS

In the **Method** section, we proposed four approaches for text rewriting using large language models (LLMs). Among them, *Back Translation*, *Summarize*, and *Rewrite* are commonly adopted in practical scenarios. These three methods support self-iterative generation, where the output from the previous round is re-used as input for the next generation. This iterative strategy aims to increase variability and richness in the rewritten content.

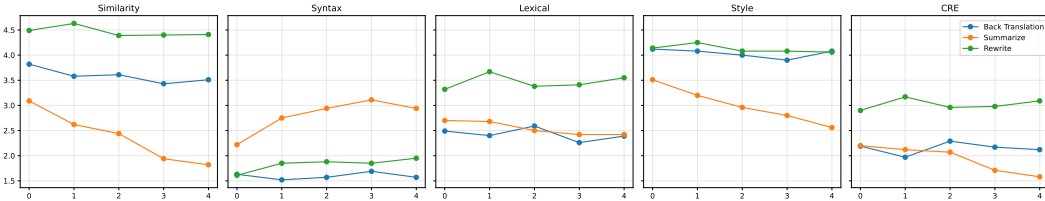

Figure 6: Performance across five dimensions over five self-iterative rounds for three LLM rewriting methods.

To evaluate their performance, we applied each method to a dataset of 100 text samples, using the Qwen2.5-32B model as the base generator. Each method underwent five rounds of self-iteration, and the outputs were scored using the QWQ-32B-Multi-Reason model across five evaluation dimensions: *Similarity*, *Syntax*, *Lexical Richness*, *Style*, and *CRE* . The results are illustrated in Figure 6.

From the figure, we observe that in the first round (iteration 0), the *Rewrite* method achieves the best performance, particularly in CRE (score of 2.90), indicating superior creative rewriting capabilities. In comparison, the scores of *Back Translation* and *Summarize* are noticeably lower, with Summarize being the weakest overall. As the number of self-iteration rounds increases, the performance of *Summarize* continues to degrade across all five metrics, suggesting that iterative summarization leads to semantic erosion and decreased quality. In contrast, both *Back Translation* and *Rewrite* show

relatively stable behavior with minor fluctuations across iterations. Notably, *Rewrite* maintains its high performance consistently throughout all five rounds, indicating it is less sensitive to degradation from iterative generation. *Back Translation*, while not as strong as *Rewrite*, also preserves reasonable quality across rounds and demonstrates better long-term stability than *Summarize*.

| Model | Sim. | Syntax | Lex. | Style | CRE | | | |
| --- | --- | --- | --- | --- | --- | --- | --- | --- |
| | | | | | QWQ | GPT5 | DeepSeek | AVG |
| Baseline | **4.63** | 2.37 | 4.05 | **4.51** | 3.41 | 3.42 | 3.72 | 3.52 |
| Pipeline $_{3\ Rounds}$ | 4.61 | 3.06 | 3.97 | 4.39 | 3.57 | **3.57** | 3.90 | 3.68 |
| Pipeline $_{5\ Rounds}$ | 4.46 | **3.65** | **4.11** | 4.41 | **3.92** | 3.56 | **4.11** | **3.86** |
| Pipeline $_{w/o\ Scoring}$ | 4.37 | 2.67 | 3.87 | 4.12 | 3.18 | 3.35 | 3.66 | 3.40 |

Table 6: Evaluation results for rewrite pipelines based on QWQ_32B. CRE is now evaluated by three different models (QWQ-32B, GPT5-mini, DeepSeek-671B-Reason) to compare their judgments.

We propose a generation pipeline guided by rewrite scores, as shown in Figure 1. Each iteration incorporates feedback from the previous rewrite, based on fine-grained scores across multiple dimensions, to indicate which aspects need improvement. This iterative feedback steers the model toward higher-quality outputs.

A natural question is whether scoring criteria (e.g., similarity, syntax, lexicon, style) should be explicitly weighted, since semantic consistency is often more critical than, say, lexical variation. However, these dimensions are not combined via a fixed-weighted formula to compute CRE. Instead, they serve as intermediate signals to guide rewriting. Changes in syntax or style often affect similarity, making predefined weights unreliable across cases. Thus, we use LLMs to produce holistic CRE judgments, implicitly balancing trade-offs among dimensions. This ensures scoring reflects overall quality rather than artificial sub-dimension weighting.

Table 6 shows the results. Starting from the strong QWQ-32B baseline, our pipeline improves CRE (AVG) from 3.52 to 3.68 in three rounds, and to 3.86 in five rounds. This indicates that even powerful LLMs benefit from structured, score-based feedback.

One may worry whether improvements in model-based scoring reflect genuine quality gains or simply better alignment with a particular scorer's bias. To mitigate this, we used multiple independent evaluators: besides QWQ-32B, we included GPT5-mini and DeepSeek-671B-Reason. As Table 6 shows, improvements are consistent across all judges, suggesting the pipeline genuinely enhances quality rather than overfitting to one model. While human evaluation would be stronger, cross-model consistency indicates gains are not due to self-judgment.

We also ablate the use of explicit score-based feedback. In Pipeline$_{w/o\ Scoring}$, we provide only vague feedback (e.g., "the previous rewrite was not good enough, please improve it"). This results in a clear performance drop (CRE = 3.40), confirming that detailed scoring is crucial for high-fidelity rewriting.

## 5 CONCLUSION

This paper proposes a structured and reliable method for evaluating sentence rewriting quality, moving beyond traditional surface-level metrics like BLEU and ROUGE. We define four key dimensions, semantic consistency, syntax variation, lexical substitution, and style fidelity, and use these to build a prompt-based evaluation framework.

Our experiments show that large language models, especially QWQ-32B with multi-dimensional scoring prompts, can match human-level judgment accuracy . We also demonstrate that score-guided rewriting improves generation quality by 9.66%, outperforming traditional methods like back translation and summarization.

Overall, our framework provides a more accurate, interpretable, and scalable approach for both evaluating and generating high-quality sentence rewrites.

# 6 ETHICS STATEMENT

This work adheres to the ICLR Code of Ethics. In this study, no human subjects or animal experimentation was involved. All datasets used were sourced in compliance with relevant usage guidelines, ensuring no violation of privacy. We have taken care to avoid any biases or discriminatory outcomes in our research process. No personally identifiable information was used, and no experiments were conducted that could raise privacy or security concerns. We are committed to maintaining transparency and integrity throughout the research process.

# 7 REPRODUCIBILITY STATEMENT

We have made every effort to ensure that the results presented in this paper are reproducible. All code and datasets have been made publicly available in an anonymous repository to facilitate replication and verification. The experimental setup, including training steps, model configurations, and hardware details, is described in detail in the paper. We will also make the complete workflow code available in the future to assist others in reproducing our experiments.

Additionally, the datasets used in this paper, including our self-constructed paraphrase scoring dataset, are publicly available, ensuring consistent and reproducible evaluation results.

We believe these measures will enable other researchers to reproduce our work and further advance the field.

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
