# OpenReview forum: "Towards Human-Preferences Chinese Rewriting Evaluation: Prompt-Based Scoring with Large Language Models"
_ICLR.cc/2026/Conference — ICLR 2026 Conference Withdrawn Submission_

### Official Review · Reviewer_2c2a · 2025-10-30

**Soundness:** 2
**Presentation:** 3
**Contribution:** 2
**Rating:** 4
**Confidence:** 5

**Summary:**

This paper proposes a prompt-based, multi-dimensional framework for evaluating and guiding Chinese sentence rewriting. The method integrates semantic consistency, syntactic variation, lexical diversity, and stylistic fidelity into a unified evaluation scheme using large language models (LLMs), particularly QWQ-32B. The authors report that their LLM-based scoring achieves a Spearman correlation (ρ = 0.6121) with human annotation, comparable to inter-human agreement (ρ = 0.6076). The study also introduces a multi-round rewrite generation pipeline, demonstrating a 9.66% improvement in rewriting quality over traditional methods such as back translation or summarization. Overall, the paper addresses a relevant gap in LLM evaluation for rewriting tasks but could benefit from deeper theoretical analysis and broader generalization.

**Strengths:**

1. The paper tackles an important and underexplored problem—evaluation of rewriting quality—by moving beyond superficial n-gram similarity toward structured, semantically grounded scoring.
2. The proposed four-dimensional evaluation criteria (semantic, syntactic, lexical, and stylistic) are well-motivated and clearly defined, providing a systematic framework for human-like assessment.
3. The empirical results are convincing, demonstrating that the QWQ-32B model aligns closely with human judgment, achieving near human-level agreement on a manually annotated dataset.
4. The proposed score-guided multi-round rewriting pipeline is a practical and well-validated contribution that improves rewrite quality across multiple evaluation models (QWQ, GPT-5, DeepSeek).

**Weaknesses:**

1. While the framework is well-engineered, it remains primarily empirical and methodological, lacking theoretical justification for how the four evaluation dimensions interact or should be weighted within the overall scoring process.
2. The current experiments focus solely on Chinese rewriting tasks, without demonstrating transferability to English or multilingual contexts, which limits the framework’s generalization and broader applicability.
3. Human–model bias analysis is somewhat shallow; incorporating qualitative case studies of where model and human evaluations diverge would help identify systematic errors or strengths in the scoring model.
4. Quantitative gains from the multi-round rewriting pipeline are promising, yet it is unclear whether these improvements reflect true semantic enhancement or overfitting to the model’s own scoring bias.
5. Prompt design exploration remains limited; analyzing why multi-reason prompting reduces performance in smaller models and how to optimize prompt granularity would provide more actionable insights.
6. Visualization and interpretability could be strengthened by presenting concrete rewriting examples, detailed score breakdowns, and side-by-side comparisons of human, BLEU, and LLM-based assessments to illustrate practical advantages.

**Questions:**

1. How robust is the scoring framework across different rewriting tasks, such as summarization or simplification, which may have distinct semantic–syntactic trade-offs?
2. Has the proposed metric been tested on out-of-domain or noisy data, such as user-generated text, to verify its stability and bias?
3. Could a hybrid metric combining QWQ-based scoring and embedding-based similarity (e.g., BERTScore) further enhance reliability?
4. How computationally intensive is the multi-round feedback pipeline, and is it scalable for large-scale rewriting datasets?

---

### Official Review · Reviewer_u3SR · 2025-10-31

**Soundness:** 2
**Presentation:** 1
**Contribution:** 3
**Rating:** 2
**Confidence:** 3

**Summary:**

This submission focusses on the evaluation of Chinese sentence rewriting. The authors compare humans and LLMs at the task of scoring rewrites with respect to 5 different criteria. The rewrites are generated using 4 different prompts.

**Strengths:**

- The 5 criteria are well-motivated
- The superiority over symbolic/heuristic metrics is demonstrated effectively

**Weaknesses:**

My largest concern is that this work seems quite detached from related work. The weaknesses of symbolic metrics in many NLP areas are well-known and learnt / neural-network-based metrics have been used for years now. For example, the main MT metrics are learnt (BLEURT, COMET, etc.), not BLEU. Some of these do not use the source sentence (e.g. BLEURT), so they could serve as baselines in this work.

Even more related, LM-as-a-judge is an active and quite large area of research, but the paper contains little reference to it (except a short note at the end of Sec. 2). This work could be a nice addition to this existing line of research, but unfortunately it does not position itself as such, and does not compare against relevant baselines from that area.

Minor comments:
- Formatting of citations: Use brackets
- Tiny figures - I feel that this has become more common recently, but imo it should be a reason for desk-rejection if there is absolutely no chance of reading it on a print-out.

**Questions:**

- How do you see your work in context of existing work around LM-as-a-judge or learnt NLP (e.g. MT) metrics?

---

### Official Review · Reviewer_FUM3 · 2025-11-01

**Soundness:** 2
**Presentation:** 3
**Contribution:** 2
**Rating:** 2
**Confidence:** 5

**Summary:**

The authors propose a prompting strategy for evaluation of the Sentence Rewriting task. As part of this research, they have collected a novel human-annotated dataset for this task, achieving moderate ~0.6 Spearman correlation between annotators. The effectiveness of the proposed approach is evaluated on the collected dataset and compared to baselines consisting of lexical overlap metrics and semantic similarity-based ones. The authors additionally propose a sentence rewriting pipeline, which incorporates the proposed evaluation approach to iteratively deliver better rewritings.

**Strengths:**

S1. The authors proposed effective prompting strategies for reasoning and non-reasoning LLMs to evaluate sentence rewriting task, focusing on Chinese language. According to the results, there is indeed a boost in evaluation quality as compared to the mentioned baselines.
S2. The authors conducted extensive experiments on models of different sizes, which allows us to draw conclusions about scaling behavior of the proposed evaluation metric.

**Weaknesses:**

W1. The related work section fails to mention a plethora of recently (and not so) published trained MT evaluation metrics, such as BLEURT, COMET, xCOMET, MetricX, GEMBA-DA/MQM/ESM or more general, such as Prometheus and M-Prometheus. Given that authors' motivation for this research is built upon flaws of existing evaluation metrics, it is a substantial weakness that they haven't discussed the most recent and most performant (as per benchmarks) of those metrics.

W2. Stemming from W1, lack of comparison of the proposed method with most recent baselines is a flaw. This paper would benefit from comparison with such baselines as Prometheus or M-Prometheus, as they allow to customize evaluation rubrics.

W3. Authors claim in L502-503, that the datasets used in this paper are publicly available, yet they are in fact not. There are no links nor access mode explanations for any datasets anywhere in the paper, including authors-collected evaluation dataset.

**Questions:**

Q1. How well does your method perform in comparison to any of the recently published metrics (both MT-specific and more general)?

---

### Official Review · Reviewer_9Fq9 · 2025-11-02

**Soundness:** 2
**Presentation:** 2
**Contribution:** 3
**Rating:** 4
**Confidence:** 4

**Summary:**

The paper suggests an LLM based evaluation strategy for sentence rewrites. Specifically, it suggest a prompt that works by addressing the issues with current evaluation metrics like BLEU, ROUGE-L, which typically do not capture semantics very robustly and word-embeddings, Sentence-BERT which don’t capture variations very robustly. The provided method evaluates across 4 particular dimensions; semantic consistency, syntactic structure, lexical variation and style length. By designing prompts for scoring with models like QWQ-32B, the authors report a high Spearman correlation with human judgments (0.6121), close to inter-annotator reliability.

Suggestions:
1\. Line 078: This is just the rewrite of the previous paragraph. No need to separate it out if one immediately follows the other.

2\. Line 094-098: Check bracketing for the citations in this section.

3\. Line 359: Provide some explanations as well about why Qwen3, LLaMA3 show weaker and inconsistent results.. Give examples where it fails, do a thorough analysis.

4\. Line 448-451: What is the generation pipeline exactly ? How does it do it? This part is not clear from the figure and this paragraph. Can you give an example and elaborate on this?

5\. Line 459-461: It makes strong sense to continue and see if and where it saturates - at what iteration. Is CRE = 5 ideal, always, or are there situations where one might not present CRE so high? Does it not then become a thresholding based situation incase CRE is not equal to 5.

**Strengths:**

1. The paper identifies some critical weaknesses in current metrics, highlighted in Table2, and proposes a prompt to mitigate those.

2. Comprehensive analysis on Chinese rewriting via multiple LLMs.

**Weaknesses:**

1. Lack of clarity: Presentation of key parts of the papers in convoluted.

2. Code link is not available, weak rationales for parameter and design choices (see summary)

3. The rationale for observed weaknesses in models like Qwen3 and LLaMA3 is insufficient; the paper points out their inconsistency but does not provide concrete examples or thorough failure analysis.

**Questions:**

1) Line 075: $\rho$ = 0.593 here but in the summary it is mentioned as 0.6121. Why is there a discrepancy?

2) Section 3.1: This section is very confusing. Is the task - rewriting semantically similar sentences or annotating them?
In line 139 (Requirement 1) it is written "express in different words.."
In line 129 it says "..experts rated"
In line 161 it is "we" re-annotated 730 samples - Who are "we"? What was the original dataset and annotation, if they are being re-annotated?

3) Line 137: What does the title of this table mean? Are you rewriting the quality scoring criteria
Also Requirement 1 asks the annotator to rewrite the sentence, does it mean that they are scoring what they are writing? Should it not be just a scoring mechanism instead given a pair of sentences ?

4) Line 424: Why was 5 chosen as the number of iterations? What was the deciding criteria?

---

### Note · Authors · 2025-11-12

I have read and agree with the venue's withdrawal policy on behalf of myself and my co-authors.